# Research on the Shock Wave Overpressure Peak Measurement Method Based on Equilateral Ternary Array

**DOI:** 10.3390/s24061860

**Published:** 2024-03-14

**Authors:** Yongjian Zhang, Peng Peng, Tao Lin, Aiwei Lou, Dahai Li, Changan Di

**Affiliations:** School of Mechanical Engineering, Nanjing University of Science and Technology, Nanjing 210094, China; yumi9621@163.com (Y.Z.); p.peng@njust.edu.cn (P.P.); ltnjust@163.com (T.L.); luoaiwei@njust.edu.cn (A.L.); lidahai@njust.edu.cn (D.L.)

**Keywords:** shock wave overpressure, shock wave velocity, dynamic measurement, measurement model, equilateral ternary array

## Abstract

The measurement process of ground shock wave overpressure is influenced by complex field conditions, leading to notable errors in peak measurements. This study introduces a novel pressure measurement model that utilizes the Rankine−Hugoniot relation and an equilateral ternary array. The research delves into examining the influence of three key parameters (array size, shock wave incidence angle, and velocity) on the precision of pressure measurement through detailed simulations. The accuracy is compared with that of a dual-sensor array under the same conditions. Static explosion tests were conducted using bare charges of 0.3 kg and 3 kg TNT to verify the numerical simulation results. The findings indicate that the equilateral ternary array shock wave pressure measurement method demonstrates a strong anti-interference capability. It effectively reduces the peak overpressure error measured directly by the shock wave pressure sensor from 17.73% to 1.25% in the test environment. Furthermore, this method allows for velocity-based measurement of shock wave overpressure peaks in all propagation direction, with a maximum measurement error of 3.59% for shock wave overpressure peaks ≤ 9.08 MPa.

## 1. Introduction

In modern warfare, terrorist attacks, and accidental explosions, the blast shock wave stands out as a key contributor to casualties in various explosive events [1,2,3,4]. Studies reveal that over 60% of injuries to the torso and head of blast victims are related to the blast shock wave [5]. Predicting, assessing, protecting against, and treating blast-related injuries require accurate data on shock wave pressure; therefore, precise measurement of shock wave pressure values is crucial. Currently, the predominant test methods for effectively acquiring the shock wave pressure parameters include the impact target method, the pressure sensor method, the velocimetry method, and so forth [6,7,8,9,10,11]. Although the effect target method is more intuitive, it cannot accurately obtain the shock wave overpressure value [12,13]; however, due to the complex and harsh conditions at the explosion test site, direct pressure sensor measurements are susceptible to additional interferences that may distort the effective signal. Furthermore, the installation state of the sensor can also impact the accuracy of the measurement results [14,15,16]. Therefore, direct measurement of accurate shock wave overpressure values using pressure sensors requires high requirements for test site and sensor installation, which requires a large cost [17,18,19]. In contrast, measuring peak overpressure using the velocity method is an indirect approach that may help mitigate the significant interference encountered with direct pressure transducer methods.

Researchers have conducted numerous studies on shock wave overpressure measurements using the velocity method, which is based on the relationship between shock wave overpressure and velocity (The Rankine−Hugoniot Equations). RG Racca and JM Dewey, among others, introduced a shock wave flow visualization test technique based on this principle that includes Refractive Image Analysis (RIA) and Particle Trajectory Analysis (PTA) for shock wave testing [20,21]. Jindrich Kucera utilized the RIA principle to examine the propagation velocity of shock wave overpressure through high-speed photography combined with a background curtain, thereby determining the overpressure value. The results were then compared with those obtained from free field sensor tests, showing better alignment with the overpressure peak [22]. Li Bin used the ripple velocity method to achieve a large equivalent blast field shock wave overpressure peak measurement, with test results demonstrating a relatively close correlation between the ripple velocity method and the pressure sensor test results [23]. Jianjun An utilized a laboratory shockwave tube to validate the viability of the pressure transducer velocity method for indirectly testing the peak overpressure of shock waves [24]. The literature [25] successfully measures the propagation velocity of underwater shock waves by installing a pressure probe inside a circular tube. The method utilizes the circular tube to direct the underwater shock wave, allowing for precise control of the pressure probe spacing with certain anti-disturbance capabilities. Nevertheless, this method could lead to local pressure disturbances and possible issues, such as probe spring jamming. To address these challenges, the literature developed an optical measurement system. The device is aligned in the direction of the shock wave propagation during testing, enabling measurement of shock wave velocity and pressure peaks by assessing plate displacement, radiation changes, and alterations in light reflection induced by the shock wave. The method is a sophisticated combination of mechanical and optical systems, posing challenges for easy portability and practical use in a field blast environment. The literature [26] analyzes the variation in shock wave front reflectivity at different shock pressures based on the Drude-free electron gas model, and consequently derives the equation for measuring shock wave velocity behind a sapphire window. This method necessitates a continuous laser enhancement throughout the measurement process, making it more challenging to implement on the test site.

In summary, shock wave velocity measurements are commonly carried out through visualization tests and pressure transducer tests. Visualization testing is typically suitable for large-scale explosion scenarios, but it imposes stringent requirements on the background environment. On the other hand, the conventional method of using a double pressure sensor for testing necessitates alignment between the direction of shock wave propagation and the orientation of sensor deployment. However, during real-time dynamic tests, it is challenging to accurately predict the blast point in advance, leading to difficulties in aligning the propagation direction with the direction of sensor deployment, which frequently leads to significant measurement errors as a consequence. To tackle these challenges, this paper proposes a novel approach to the measurement of explosion shock wave overpressure. The proposed method utilizes an equilateral ternary array for enhanced precision in measurements. This method is based on a piezoelectric pressure sensors, thereby effectively circumventing the stringent requirements of visual measurement on the measurement background. By leveraging the advantages of a triangular array, it effectively mitigates the impacts of the incident angle of shock waves in indirect measurement processes. Furthermore, this method only requires measuring the overpressure arrival time to determine pressure peak values, thereby effectively avoiding the impacts of mechanical shock, thermal shock, electromagnetic shock, and installation condition restrictions on the measurement accuracy, which typically manifest in sensor-based direct measurement methods.

## 2. Shock Wave Overpressure Peak Measurement Model

### 2.1. Basic Properties of Shock Waves and State Parameter Equations

During the process of an explosion, the physical parameters experience abrupt changes both in front of and behind the shock wave front. Due to the high propagation velocity of the wave, the propagation process can be viewed as an adiabatic process. Consequently, this satisfies the conditions for using the Rankine−Hugoniot relations [27].

When a shock wave propagates in still air, as shown in Figure 1a, its propagation velocity of D. $P_{0}$, $\rho_{0}$, $\nu_{0}$, $e_{0}$ is denoted by the pre-wave gas state parameters pressure, density, velocity and energy. As shown in Figure 1b, P, ρ, ν, e represents the post-wave gas state parameters, which include pressure, density, velocity, and energy. Therefore, based on the law of conservation of mass, the rate of mass inflow per unit time within a specific volume is equal to the rate of mass outflow, as depicted in Equation (1).
(1)$$\rho_{0}(D-\nu_{0})=\rho (D-\nu )$$

According to the law of conservation of momentum, the rate of change in momentum exerted on the medium during the propagation of the shock wave is equal to the impulse of the applied force, as depicted in Equation (2).
(2)$$P-P_{0}=\rho_{0}(D-\nu_{0})(\nu -\nu_{0})$$

According to the law of conservation of energy, the change in energy within the system is equal to the work performed by the external force. Within a unit time, when the undisturbed medium unit mass of internal energy is $e_{0}$, the kinetic energy flowing into the wavefront surface is represented by $\frac{1}{2}\rho_{0}{\left(D-\nu_{0}\right)}^{2}$. The kinetic energy flowing out is denoted as $\frac{1}{2}\rho {\left(D-\nu \right)}^{2}$. The work performed by the pressure on both sides of the wavefront surface is indicated by $P_{0}(D-\nu_{0}),P(D-\nu )$. It can be derived from Equation (3).
(3)$$e-e_{0}=\frac{P\nu -P_{0}\nu_{0}}{\rho_{0}(D-\nu_{0})}-\frac{1}{2}(\nu^{2}-{\nu_{0}}^{2})$$

### 2.2. Measurement Model

For a multi-party gas, the equation of state is shown in Equation (4).
(4)$$e=C_{\bar{\nu }}T=\frac{1}{\gamma -1}P\bar{\nu }$$

In Equation (4), $\bar{\nu }$ represents the specific volume; γ denotes the specific heat capacity ratio, and under normal conditions, γ = 1.4.

By combining Equations (1)–(4), Equation (5) can be deduced as:(5)$$P-P_{0}=\frac{2}{\gamma +1}\rho_{0}{\left(D-\nu \right)}^{2}\left[1-\frac{\gamma P_{0}/\rho_{0}}{{\left(D-\nu_{0}\right)}^{2}}\right]$$

Assuming the speed of sound in the medium is $c_{0}=\sqrt{\gamma \frac{P_{0}}{\rho_{0}}}$ that the Mach number of the shock wave is $M=\frac{\nu -\nu_{0}}{c_{0}}$. With Equation (5) in mind, then the shock wave pressure value and Mach number relationship (the Rankine−Hugoniot relation) can be obtained as in Equation (6). Therefore, the shock wave overpressure can be calculated simply by measuring the shock wave propagation velocity at the measurement point.
(6)$$P=\frac{2\gamma }{\gamma +1}\left(M^{2}-1\right)P_{0}$$

This paper suggests using an equilateral ternary array for measuring the shock wave propagation velocity. As shown in Figure 2, the test platform is located on the coordinate system (x,y) with the center at the point *O*, *S*_1_($\frac{L}{2}$,$\frac{\sqrt{3}}{2}L$), *S*_2_(*L*,0), *S*_3_(0,0) representing the pressure sensor locations forming the equilateral triangular velocity array, while *S*_0_($\frac{L}{2}$,$\frac{\sqrt{3}}{6}L$) is situated at the center of the circular test platform, coinciding with the center of triangle Δ*S*_1_*S*_2_*S*_3_. The attenuation of the shock wave in a small local area is approximately linear, so the shock wave velocity obtained by the equilateral triangular array can represent the shock wave velocity at S_0_.

Assume that the sensors are arranged in an equilateral ternary array *S*_1_*S*_2_*S*_3_ with side length *L*, while the shock wave propagates at an angle *β* (shock wave incidence angle) relative to *S*_1_*S*_0_ with a propagation velocity of *v*. The time interval *t*_12_ and *t*_13_ between the arrival of the shock wave at the sensors *S*_1_*S*_2_ and *S*_1_*S*_3_ is then shown in Equation (7).
(7)$$\left\{ \begin{array}{l} t_{12}=\frac{L*\cos\left(\beta +30{}^{\circ}\right)}{v}\\ t_{13}=\frac{L*\cos\left(\beta -30{}^{\circ}\right)}{v} \end{array}\right.$$

According to Equation (7), the relationship between the shock wave propagation velocity v and L, t_12_, t_13_ can be deduced as shown in Equation (8).
(8)$$\nu =\frac{\sqrt{3}L}{2\sqrt{t_{12}^{2}+t_{13}^{2}-t_{12}\times t_{13}}}$$

By combining Equations (6) and (8), Equation (9) is derived for calculating the peak pressure of shock waves based on the equilateral ternary array.
(9)$$P=\frac{2\gamma }{\gamma +1}*\left[{\left(\frac{\sqrt{3}L}{2\sqrt{\left(t_{12}^{2}+t_{13}^{2}-t_{12}\times t_{13}\right)}}-v_{0}\right)}^{2}/c_{0}^{2}-1\right]*P_{0}$$

From Equation (9), it can be seen that the equilateral ternary array shock wave pressure peak measurement is related to *L*, *t*_12_, *t*_13_, $v_{0}$, γ, $c_{0}$ and $P_{0}$, where γ, $c_{0}$, $P_{0}$ are constants under known environmental conditions and $v_{0}$ is ambient air velocity, which can generally be obtained directly from environmental monitoring values.

## 3. Analysis of Key Parameters

### 3.1. Simulation

Based on Equation (9), it is evident that, apart from the parameters influenced by environmental factors, the calculation of peak overpressure is primarily determined by *L*, *t*_12_ and *t*_13_. The errors introduced by *t*_12_ and *t*_13_ mainly originate from the response output of the sensor, as well as the signal conditioning circuit filtering, amplification, shaping, and time-difference extraction processes. The principle of arrival time measurement error is illustrated in Figure 3.

In Figure 3, for the amplified shock wave signal, *V*_max_ is the maximum amplitude, *V*_min_ is the minimum amplitude, *V*_ref_ is the judgment threshold level, *τ*_max_ is the maximum rising edge duration, *τ*_min_ is the minimum rising edge duration, *t*_max_ is the maximum time when the threshold level is triggered, and *t*_min_ is the minimum time when the threshold level is triggered. For the same shock wave signal after the signal acquisition and processing process, the maximum measurement time error Δ*t*_max_ = *t*_max_ − *t*_min_.

Combined with Equation (7), the actual time interval between the arrival of the shock wave at the array sensor can be obtained as in Equation (10).
(10)$$\left\{ \begin{array}{l} t_{12}^{\prime }=\frac{L*\cos\left(\beta +30{}^{\circ}\right)}{v}+\Delta t_{12}\\ t_{13}^{\prime }=\frac{L*\cos\left(\beta -30{}^{\circ}\right)}{v}+\Delta t_{13} \end{array}\right.$$

Δ*t*_12_ represents the reading error of the time interval when the shock wave passes over sensors 1 and 2, while

Δ*t*_13_ represents the reading error of the time interval when the shock wave passes over sensors 1 and 3.

By combining Equations (9) and (10), it can be demonstrated that the error of the pressure measurement model in this paper is primarily influenced by the array edge length *L*, the shock wave incidence angle *β*, the shock wave velocity *v*, and the measurement time error Δ*t*. The sensors used for the validation of this study were the 603C series piezoelectric pressure sensors from Kistler and the FE-408 data collector from Elsys. The rising time of the 603C sensor is less than 0.4 microseconds, and the time resolution of the FE-408 data acquisition instrument reaches a maximum of 0.05 μs. Therefore, the overall error in rising time determination can be effectively controlled to within ǀΔ*t*ǀ ≤ 0.5 μs.

A numerical simulation is conducted to evaluate the impact of *L*, *β*, and *v* on the precision of measuring shock wave overpressure peaks using the equilateral ternary array. The numerical simulation parameters are set as shown in Table 1. The measurement time error is taken as the maximum value Δ*t* = Δ*t*_12_ = Δ*t*_13_ = 0.5 μs. The shock wave velocity ‘*v*’ ranges from 400 m/s to 7000 m/s (corresponding to shock wave pressures of 0.045 MPa to 49.66 Mpa). The length of the array, *L*, varies from 0.05 m to 0.2 m. Considering the symmetry of the array, the angle of incidence, *β*, is selected from the range of −60° to 60°. The numerical simulation results are displayed in Figure 4, providing a general overview. It can be observed that, as *L* increases, the measurement error decreases. However, the rate at which the error decreases gradually slows down with the increasing *L*. Near *β* = 0°, the measurement error is minimal, while on both sides of 0°, the measurement error increases with the increase in *β*. With the exception of *v* = 400 m/s, the measurement error increases as *v* grows. Moreover, the trend of the measurement error increasing gradually intensifies with higher *v* values.

### 3.2. Analysis

#### 3.2.1. Influence of Array Size

In order to assess the impact of *L* on the accuracy of the pressure measurement method based on an equilateral ternary array, the numerical simulation results are analyzed with Δ*t* = 0.5 μs, *β* = 0°, *v* = 400 m/s to 7000 m/s., and *L* ranging from 0.05 m to 0.2 m. The numerical simulation results are also compared with those of the dual-sensor array under the same conditions. The analysis of Figure 5 indicates that the measurement error decreases as *L* increases, with this being particularly evident when *v* > 3000 m/s and L is larger. However, the pattern of error reduction becomes more stable at *L* ≥ 0.15 m, with minimal contribution to enhancing the accuracy of overpressure measurement. For instance, at *L* = 0.15 m, *v* = 3000 m/s, the error is −0.02%, and at *v* = 7000 m/s, the error is −0.04%. The dual-sensor velocity method of pressure measurement error also exhibits a decreasing error trend with increasing *L*, but the overall error is higher. For instance, at *L* = 0.15 m, *v* = 3000 m/s, the error is −2.0%, while at *v* = 7000 m/s, the error is −4.52%. Under identical conditions under the same *L*, the measurement error is significantly greater than that of the equilateral ternary array.

In summary, *L* = 0.15 m is better when *v* = 400 m/s to 7000 m/s (corresponding to overpressure is 0.0454 MPa to 49.66 MPa).

#### 3.2.2. Influence of Shock Wave Incidence Angle

In order to evaluate the degree of influence of *β* on the accuracy of the pressure measurement method based on an equilateral ternary array, the numerical simulation results are analyzed with Δ*t* = 0.5 μs, *L* = 0.15 m, *v* = 400 m/s to 7000 m/s, and *β* ranging from −60° to 60°. From Figure 6, it can be observed that the error is minimized at *β* = 0°. As the swing angle deviates from 0° in either direction, the measurement error starts to increase, displaying a symmetrical trend around the origin, with the maximum measurement error occurring at *β* = 60°. At *v* > 3000 m/s, the measurement error shows a significant increase with varying values of *β*. At *v* = 3000 m/s, the error of *β* = 10° is 0.67%, and the error of *β* = 30° is 2.03%. At *v* = 7000 m/s, the error of *β* = 10° is 1.43%, and the error of *β* = 30° is 4.67%. The dual-sensor velocity method of pressure measurement error is notably affected by *β*, with a sharp rise in measurement error as *β* increases. Interestingly, there is a negative correlation between the measurement error and shock wave velocity—the error decreases as the velocity increases. For example, under the condition of *v* = 3000 m/s, the measurement error is 3.15%, 33.77%, and 303.9% when *β* = 10°, 30° and 60°, respectively.

In summary, (1) the triangular array method can achieve higher accuracy in peak pressure measurement regardless of the incidence angle of the shock wave, and (2) the shock wave incidence angle changes on the equilateral ternary array measurement accuracy are significantly less important compared to the dual-sensor array.

#### 3.2.3. Influence of Shock Wave Propagation Velocity

In order to evaluate the degree of influence of *v* on the accuracy of the pressure measurement method based on an equilateral ternary array, the numerical simulation results are analyzed with Δ*t* = 0.5 μs, *β* = 0°, *L* = 0.15 m, and *v* ranging from 400 m/s to 7000 m/s. The numerical simulation results are also compared with those of the dual-sensor array under the same conditions. From Figure 7, it is evident that variations in *v* will lead to different pressure measurement errors, which increase as *v* increases. There is an inflection point in the trend of *v*’s impact on pressure measurement error at around 500 m/s. However, *v* is less than this inflection point, and the degree of error affected is significantly lower than the measurement results at high *v*. At *L* = 0.15 m, using the equilateral ternary array shock wave pressure measurement method, the maximum impact of shock wave velocity on the pressure measurement error is only −0.22%, while the maximum impact of shock wave velocity on the error reaches −4.52% when the dual-sensor array method is used for pressure measurement.

In conclusion, the equilateral triangular array velocity method can achieve a relatively high precision measurement of shock wave overpressure peak within the range of *v* = 400 m/s to 7000 m/s with significantly better measurement accuracy than that of the dual sensor array.

## 4. Test Verification

Tests were conducted using three sets of columnar TNT bare charges, with each set consisting of 0.3 kg and 3 kg charges. The height of the charges from the ground was 1.5 m, with detonation initiated at the center of the charges. The ground medium was concrete. As shown in Figure 8a, the test platform is installed at 1.8 m from the projection point of the center of the explosion. The relationship between the test platform and the direction of shock wave propagation is group 1 *β* = 0°, group 2 *β* = 10°, group 3 *β* = 30°. The test system is shown in Figure 8b and consists of a test platform, pressure sensor, a data acquisition device, and a computer. The test plate is a steel disc with a radius R = 0.15 m in which multiple pressure sensors are mounted. The pressure sensors S_1_, S_2_ and S_3_ form an equilateral ternary array of L = 0.15 m, the center of which coincides with the center of the test platform. S_1_ and S_C_ form a dual-sensor array. Meanwhile, standard pressure sensors S_0–1_ and S_0–2_ are selected at the center of the test plate to measure the reference pressure value. For convenience, all test sensors on the test platform use the 603C piezoelectric pressure sensor from Kistler, which has a high response rate and high acquisition accuracy to meet the test requirements. The main parameters of the sensor are shown in Table 2. The FE 408 Data Acquisition Device, produced by the Elsys company, enables simultaneous acquisition of 32 channels with a maximum sampling rate of up to 20 MHz and a resolution of 14 bits, meeting the test requirements for blast field shock wave pressure. The main technical parameters of the FE 408 are shown in Table 3.

The shock wave pressure curve for the 0.3 kg TNT bare charge test is shown in Figure 9, and the shock wave pressure curve for 3 kg is shown in Figure 10. From the pressure time curve, measured by the sensors at different positions on the test platform, it can be seen that the surface-reflected pressure time curve has a certain amount of noise, which is mainly caused by the explosion-induced shock vibration, thermal shock, etc., resulting in parasitic signal output from the sensor. However, it can be seen that the shock wave overpressure waveform has the characteristics of rapid rise and obvious arrival moment, which can well-meet the data analysis requirements.

The shock wave pressure time curve obtained from the test was extracted numerically, and the arrival moment differences *T*_12_, *T*_13_, *T*_1C_ of sensors *S*_1_*S*_2_, *S*_1_*S*_3_, *S*_1_*S*_C_ on the medium test platform, deviation ΔP of the disturbed direct measurement value, and the reference shock wave overpressure peak P_STD_ were obtained for the two sets of tests. The equilateral ternary array overpressure measurement value is P_3v_. The overpressure measurement value for the dual-sensor array is denoted as P_2v_, with the measurement error represented by σ (using P_STD_ as the reference true value). Detailed data is presented in Table 4. Based on the information in Table 4, the following conclusions can be drawn.

(1) In the data curves of the 0.3 kg and 3 kg TNT bare charge tests, the average error of the overpressure peak reading due to interference in the directly measured curve is 17.73%, while the average deviation of the overpressure peak obtained through the triangular array pressure measurement method is only 1.25%. This indicates that the triangular array pressure measurement method possesses a significantly superior anti-interference capability compared to the direct measurement method.

(2) The equilateral ternary array velocity method at *β* = 0°, 10°, 30° gives the average deviations as 0.5%, 0.97%, 1.55% for 0.3 kg TNT bare charge, and 1.36%, 1.65%, 1.46% for 3 kg TNT bare charge, respectively. It can be seen that the overpressure error measured by the equilateral ternary array is small and positively related to the variation in *β*. The overall variation is minimal and the errors are all insignificant. The average deviations of the dual-sensor velocimetry method under the same conditions were 2.49%, 6.8%, 47.42% and 3.25%, 5.27%, 50.45%, respectively. The measurement error exhibits an obvious positive correlation with the variation in *β*, which is significantly larger than the measurement error observed with the triangular array method. It can be concluded that the accuracy of the triangular array pressure measurement method is significantly less affected by variations in the shock wave incidence angle compared to the dual-sensor array method. It is also evident that the measurement error of shock wave overpressure under the 3 kg condition is slightly larger than the measurement error of 0.3 kg. In other words, as the shock wave velocity increases, the measurement error also increases, aligning with the numerical simulation rule.

From the test results, it is apparent that the error of the measured value is slightly larger than the numerical simulated value. This discrepancy may stem from factors such as the installation of the test plate at the test site, sensor positioning, and limitations in the error control of the standard pressure sensor, which prevent the achievement of ideal conditions. Nevertheless, the overall results show that the test measurements are in good agreement with the numerical simulated values; therefore, the validity of the numerical simulation of the parameters of the equilateral ternary array pressure measurement model is verified.

## 5. Conclusions

(1) The measurement accuracy of the equilateral ternary array shock wave pressure measurement method improves as the array edge length increases. This implies that there is a positive correlation between the measurement accuracy and the array edge length. However, when *L* ≥ 0.15 m, the trend of the measurement accuracy increasing with *L* becomes less apparent; therefore, it is advisable to select *L* = 0.15 m for the triangular array.

(2) Based on the test results, it can be observed that in the complex explosion field measurement environment, the predominant interference signal is amplitude noise. This noise does not significantly impact the measurement of shock wave arrival time. Hence, utilizing the triangular array pressure measurement method offers enhanced anti-interference capability compared to the conventional direct pressure measurement method. Furthermore, this method ensures that the measurement accuracy remains unaffected by sensor linearity, repeatability, and other characteristic parameters.

(3) According to numerical simulation and test results, it has been noted that the accuracy of pressure measurement using the dual-sensor velocity method is significantly influenced by two factors: the propagation velocity of the shock wave and the variation in the incidence angle. Although the accuracy of the triangular array pressure measurement method is also affected by the shock wave propagation velocity and shows a positive trend with changes in the incidence angle, within the shock wave propagation velocity *v* = 3000 m/s (overpressure = 9.08 MPa), the maximum induced measurement error (due to changes in the shock wave incidence angle) is 3.59%. The accuracy can basically meet the requirements of measurement accuracy in complex explosion field environments, i.e., the shock wave overpressure measurement within overpressure ≤ 9.08 MPa can basically ignore the change in the incident angle of the shock wave.

In summary, the utilization of the equilateral ternary array pressure measurement method can address the challenges associated with the direct measurement approach of traditional pressure sensors in terms of measuring shock wave pressure under severe mechanical shock, thermal shock, electromagnetic shock, and installation constraints. This method enables accurate measurement of shock wave overpressure amplitude, overcoming the limitations imposed by the aforementioned factors. Additionally, it addresses the challenge of pressure measurement accuracy inherent in the dual-sensor velocity method, which is significantly influenced by variations in shock wave propagation velocity and incidence angle. Ultimately, this approach facilitates high-precision measurement of the peak overpressure of shock waves in any given propagation direction.

## Figures and Tables

**Figure 1 sensors-24-01860-f001:**
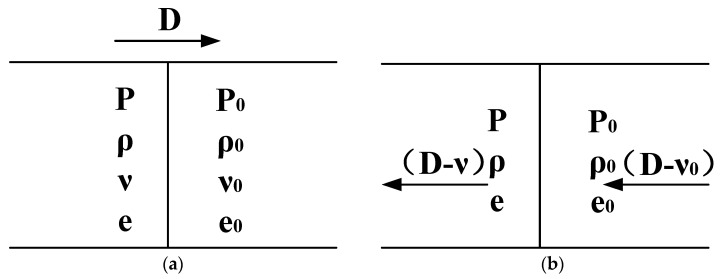
The physical parameters before and after the shock wave. (**a**) The pre-wave gas state. (**b**) The post-wave gas state.

**Figure 2 sensors-24-01860-f002:**
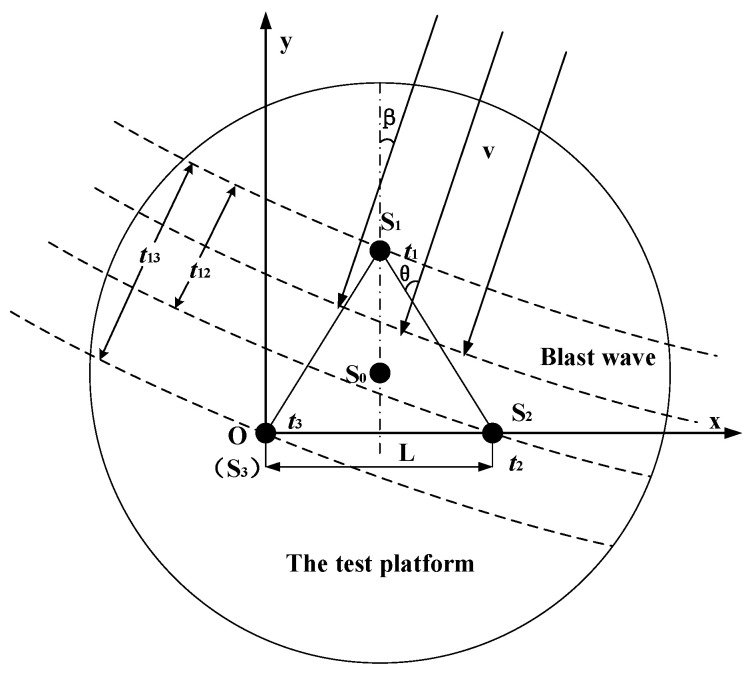
Principle of pressure measurement using an equilateral ternary array.

**Figure 3 sensors-24-01860-f003:**
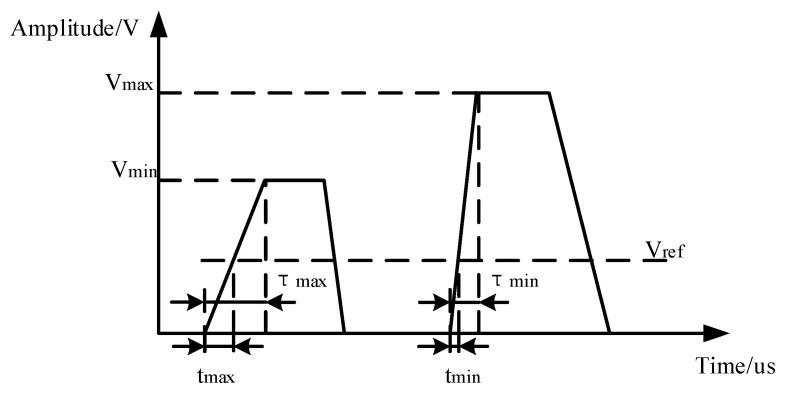
The principle of arrival time-difference measurement error.

**Figure 4 sensors-24-01860-f004:**
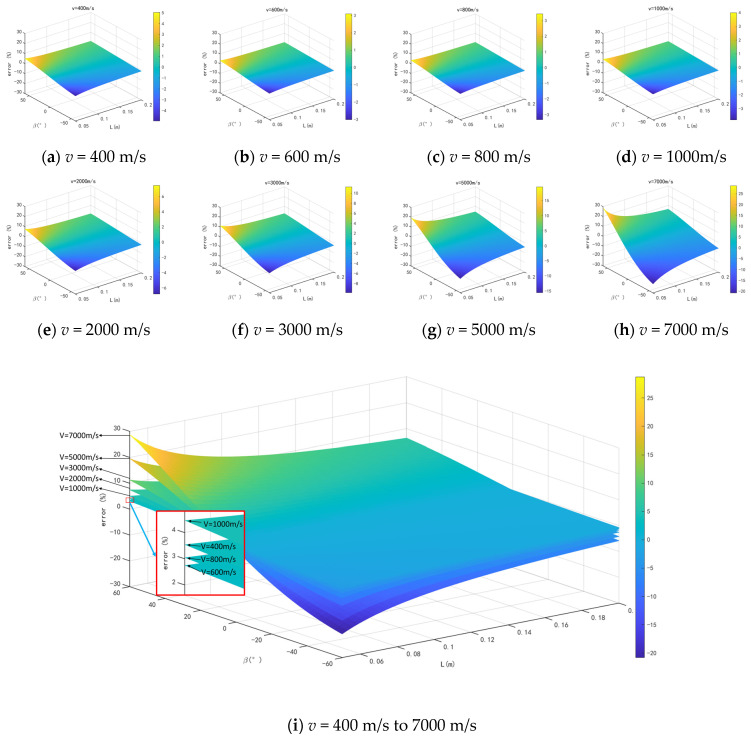
Numerical simulation results.

**Figure 5 sensors-24-01860-f005:**
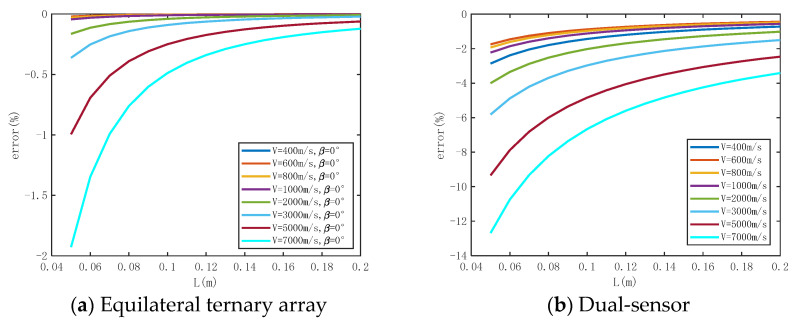
The numerical simulation results of influence of array size on measurement error.

**Figure 6 sensors-24-01860-f006:**
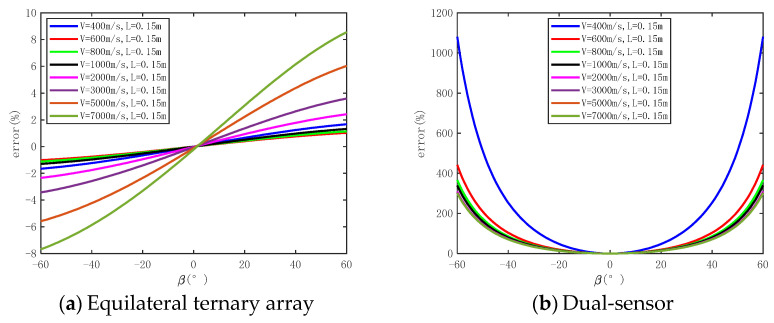
The numerical simulation results of influence of shock wave incidence angle on measurement error.

**Figure 7 sensors-24-01860-f007:**
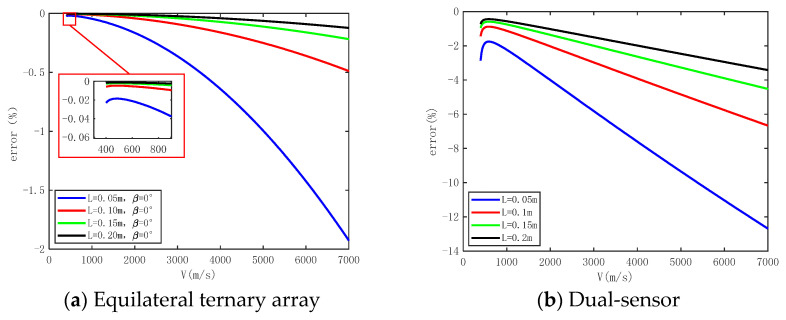
The numerical simulation results of influence of shock wave propagation velocity on measurement error.

**Figure 8 sensors-24-01860-f008:**
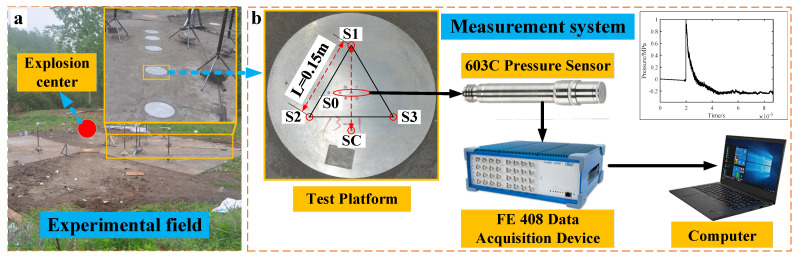
Measurement system and experiment field. (**a**) Experimental field. (**b**) Measurement system.

**Figure 9 sensors-24-01860-f009:**
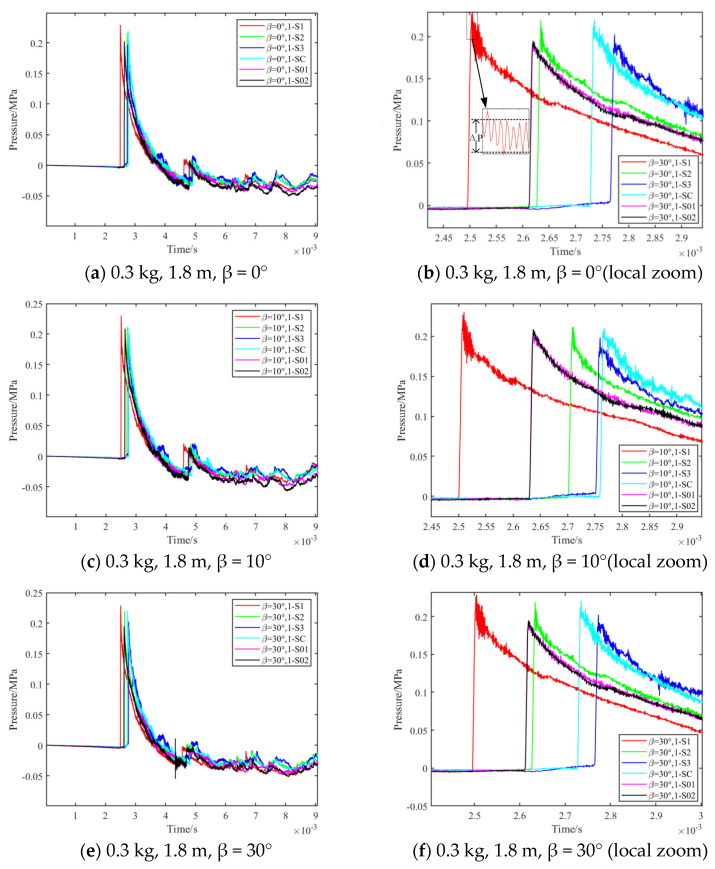
0.3 kg TNT bare charge test results.

**Figure 10 sensors-24-01860-f010:**
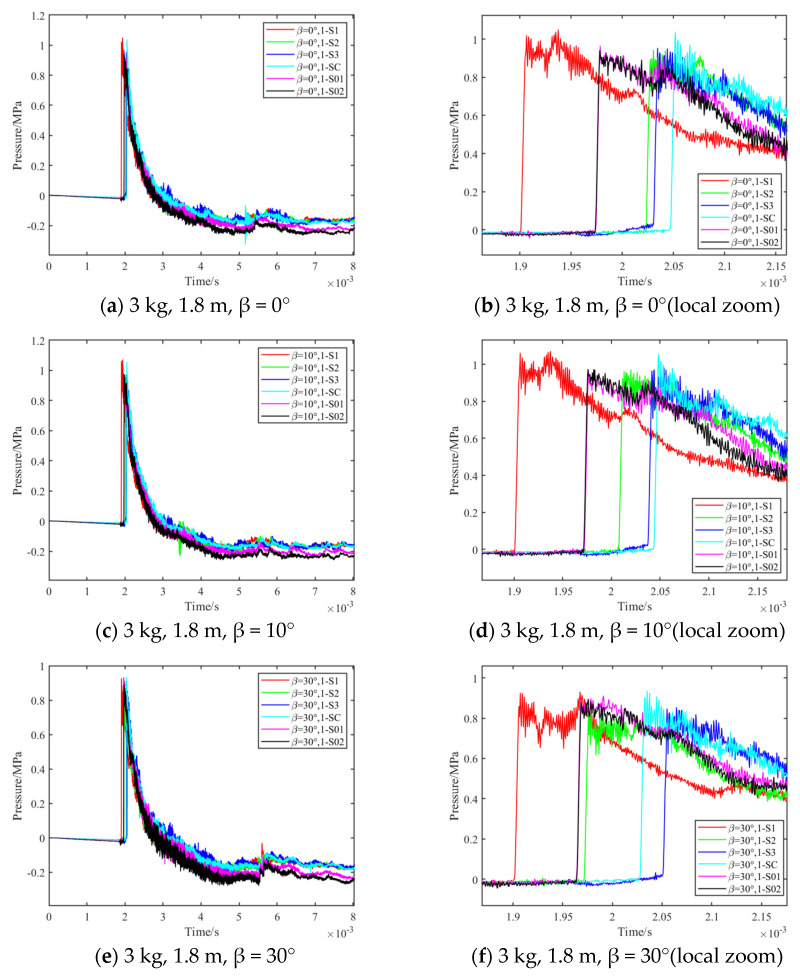
3 kg TNT bare charge test results.

**Table 1 sensors-24-01860-t001:** Numerical simulation parameter.

Parameter	Δ*t*/μs	*L*/m	*β*/°	*v*/(m/s)
Value	0.5	0.05 to 0.2	−60 to 60	400 to 7000

**Table 2 sensors-24-01860-t002:** 603C piezoelectric pressure sensor performance indicators.

Indicator Name	Technical Indicators Parameters
Maximal linearity(%FSO)	≤±1.0
Pressure range	0 … 1000 bar
Rise time (10 … 90%)	<0.4 μs
Natural frequency	>500 kHz
Mounting size	5.5 mm

**Table 3 sensors-24-01860-t003:** FE 408 Data Acquisition Device performance indicators.

Indicator Name	Technical Indicators Parameters
channels number	32 channel
input mode	ICP (IEPE), DC, AC
Maximum sampling rate	20 MS/s
Resolution	14/16-bit
Input Ranges	±100 mV, …, ±25 V, Offset Settings: 0–100%
bandwidth	10 MHz
input noise (1 MS/s)	<0.03 mVrms

**Table 4 sensors-24-01860-t004:** Test data results statistics; *L* = 0.15 m, *H* = 1.8 m (Projection distance between the explosion center and the center of the test platform), P_STD_ = (P_s0–1_ + P_s0–2_)/2, ơ_3v_ = ǀ(P_3v_ − P_STD_)/P_STD_ǀ × 100%, ơ_2v_ = ǀ(P_2v_ − P_STD_)/P_STD_ǀ × 100%, ơ_Δ_ = ǀ(ΔP − P_STD_)/P_STD_ ǀ × 100%.

TNT	*β*	T_12_	T_13_	T_1C_	P_3*v*_	P_2*v*_	ΔP	P_STD_	ơ_3*v*_	ơ_2*v*_	ơ_Δ_
kg	°	μs			MPa						
0.3	0	224.7	238.7	270.4	0.2	0.206	0.034	0.201	0.5%	2.49%	16.92%
	10	200.5	251.2	258	0.208	0.22	0.037	0.206	0.97%	6.8%	17.96%
	30	130.9	270	238.7	0.197	0.286	0.036	0.194	1.55%	47.42%	18.56%
	AVG	-	-	-	0.202	0.237	0.036	0.200	1.01%	18.9%	17.81%
3	0	122	129.9	145	0.966	0.984	0.156	0.953	1.36%	3.25%	16.37%
	10	107.4	137.4	142.2	0.984	1.019	0.171	0.968	1.65%	5.27%	17.67%
	30	70.4	150	125.7	0.903	1.339	0.168	0.89	1.46%	50.45%	18.88%
	AVG	-	-	-	0.951	1.114	0.165	0.937	1.49%	19.66%	17.64%
	overall AVG	-	-	-	-	-	-	-	1.25%	19.28%	17.73%

## Data Availability

Data are contained within the article.

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
