# Peer review of "Research on the Shock Wave Overpressure Peak Measurement Method Based on Equilateral Ternary Array"

_sensors, 2024, doi:10.3390/s24061860_

Round 1
Reviewer 1 Report
Comments and Suggestions for Authors
- In the abstract / keywords : 2 words in bolt that shouldn't - lines 7 and 23
- Commas in the wrong direction in lines :119 and 121, 140,163
- problems of figure references in the text, in lines : 119,162,199, 250, 272, 299, 322, 326
- In figure 1 : add a) and b) to help understanding of the configurations before and after the shock waves
- Line 167 : you mention "the attenuation of the shock wave in a small local area is approximately linear". Can you justify this assertion ? Refer to a paper?
- Figure 2 : angle theta not relevant, can be removed. Can you add on the figure the t12 and t13, as well of v parameters, to help understanding their geometric determination?
- Lines 178-179 : sentence lacks a verb
- For formula 8 : can you detail a bit more the way to obtain this formula ?
- Line 187 : a comma after P0 is needed instead of a dot
- Line 189 : you mention "ambiant wind speeds", isn't the right term velocity?
- For equations 10 : can you define Dt12 and Dt13, regarding your definition of Dtmax?
- Line 220 : the FE-408 data collector seems to be from Elsys company instead of Kistler
- Line 229 : to be more specific, I should have written Dt12 = Dt13 = 0.5 µs
- Figure 4 : can gain in understanding if the velocity values appear in the legend instead of the figure itself. Moreover, having the same color amplitude bar on all the 8 simulations would help in the comparison of the different results, and you should gain some space to enlarge the 3D figures and only gives the colorbar legend once.
- Line 247 : should have replaced "it is taken" by "the numerical simulation resultats of [...] is considered"
- Figure 5 : should have been easier to read and compare both geometries of sensors if using in a) and b) the same color for the same velocity considered
- Line 271 : should have replaced "it is taken" by "the numerical simulation resultats of [...] is considered"
- Line 286-290 : cut the sentence in 2, before "Any".
Replace "is significantly less than" by "are significantly less important than"
Add "by the" after used in the sentence "can be used triangular array method"
- Line 297 : should have replaced "it is taken" by "the numerical simulation resultats of [...] is considered"
- Line 319-321 : lacks a verb and should be modified
- Line 336 : Kistler needs a capital letter
- Line 339 : the FE-408 data collector seems to be from Elsys company instead of Kistler
- Legend of figure 8 : add a) and b) in the sentence.
- Table line 346 : If you choose to make a unique table for the data acquisition, remove the term "table 1" that appears in te FE-408 part and change the legend of the table.
Be carefull : in the present legend, the reference sensor doesn't appear "C" instead of "603C"
- In the table : for Maximal linearity , add "%FSO" on the line
for Natural Frequency, add ">" 500 kHz
- line 351 : the term "sensor" needs the plural
- line 366 : the sentence needs a verb
- line 376 : should have added the verb "gives" after the values of Bêta
- line 380 : should have replaced "weakly" by "small"
- lines 433-438 : sentence not clear, should be modified
- line 442 : "Angle" can be written "angle"
-in reference 9, there in a "[J]" that appears at the end of the title paper
In the test verification paragraph, it should be interested to compare your methodology with the pressure value, or the shock wave velocity determination with pressure pencil probes. We can see in the figure 8 a) that such devices were positioned on the experimental tests
In the conclusion paragraph, it should be interesting to compare the error measurements obtained with your technique regarding the Bêta angle, with other works based on pencil probes misalignements that can be found in the literature.
Author Response
Thank you very much for taking the time out of your busy schedule to provide me with these valuable suggestions. Each suggestion is of great value to my study, offering scientific and indispensable guidance for my thesis and work. Based on the suggestions you provided, I have carefully made the necessary revisions. Please see the attachment.

Reviewer 2 Report
Comments and Suggestions for Authors
It's a very interesting topics.
1. Line 178: I read through that paragraph twice and still can't get a clear picture what is the error defined. On line 182, a left bracket is given with no right bracket !! Again, a clear description must be given what is the error. This goes to the end of all the comparisons.
2. Line 199, the reference is missing. This go to the end of all article.
3. Please check Line 227 to 230. It's paradox,
Based on these points, the format must be improved for reviewing.
Comments on the Quality of English LanguageSome English writing error must be corrected for easy understanding.
Author Response
Thank you very much for taking the time out of your busy schedule to provide me with these valuable suggestions. Each suggestion is of great value to my study, offering scientific and indispensable guidance for my thesis and work. Based on the suggestions you provided, I have carefully made the necessary revisions.
- Line 178: I read through that paragraph twice and still can't get a clear picture what is the error defined. On line 182, a left bracket is given with no right bracket !! Again, a clear description must be given what is the error. This goes to the end of all the comparisons.
Thank you for your careful review. I apologize for my inadequate expression, the adjustments have been made according to your suggestions, please review.
- Line 199, the reference is missing. This go to the end of all article.
The citations have been updated as requested, please review.
- Please check Line 227 to 230. It's paradox,
I apologize for my poor expression, the paragraph has been adjusted and revised.
Based on these points, the format must be improved for reviewing.
The entire text has been proofread and revised according to your suggestions, please review.
Reviewer 3 Report
Comments and Suggestions for Authors
The text of the manuscript should be revised. It's difficult to read.
There are many errors in the text of the article due to the lack of source designation “Error! Reference source not found”, also strange commas, table 4 is unreadable, etc.
The references in the article are indicated in Latin numerals, which is inconvenient to read.
Line 110, error in the kinetic energy formula.
Author Response
Thank you very much for taking the time out of your busy schedule to provide me with these valuable suggestions. Each suggestion is of great value to my study, offering scientific and indispensable guidance for my thesis and work. Based on the suggestions you provided, I have carefully made the necessary revisions.
The text of the manuscript should be revised. It's difficult to read.
I apologize for my poor language expression. The entire text has been proofread and adjusted according to your feedback, please review.
There are many errors in the text of the article due to the lack of source designation “Error! Reference source not found”, also strange commas, table 4 is unreadable, etc.
All modifications have been made according to your suggestions, please review.
The numbering of references in the chapter has been changed to Arabic numerals, please review.
All modifications have been made according to your suggestions, the numbering of references in the chapter has been changed to Arabic numerals, please review.
Line 110, error in the kinetic energy formula.
Thank you for your reminder, the formula error has been corrected, please review.
Reviewer 4 Report
Comments and Suggestions for Authors
please see the attached file

Help from a native speaker is needed.
Author Response

(The authors gave the same response as above.)

Round 2
Reviewer 1 Report
Comments and Suggestions for Authors
Thanks for taking into account my remarks and for the complementary elements given to better evaluate your work. I have some few small modifications to ask :
- line 6 : in the abstract the first word "the" is in bold, and it should not
- line 19 : in the keywords the first word "shock" is in bold, and it should not
- line 91 : there is a line break within the middle of the sentence (to be removed)
- lines 95, 96 and 112 : these coma are still in the wrong direction
- line 214 : "equilaeral" > a "t" is missing
- line 222 : "resultats" should be replaced by "results" (I'm sorry, it is a mistake I've done in my previous comments)
I wish you success in your thesis.
Author Response
- line 6 : in the abstract the first word "the" is in bold, and it should not
Thank you for your careful review. I have made the modifications as per your suggestions.
- line 19 : in the keywords the first word "shock" is in bold, and it should not
I apologize for my carelessness. Thank you for the feedback, and I have made the necessary corrections.
- line 91 : there is a line break within the middle of the sentence (to be removed)
I have made the specific revisions as suggested. Please review them.
- lines 95, 96 and 112 : these coma are still in the wrong direction
Thank you for the suggestion. I have made the modifications in the document.
- line 214 : "equilaeral" > a "t" is missing
I apologize for the spelling errors, and thank you for your careful revisions. I have corrected them now.
- line 222 : "resultats" should be replaced by "results" (I'm sorry, it is a mistake I've done in my previous comments)
I have made the necessary corrections. Please review them.
Thank you once again for your careful proofreading. I have reviewed the entire text based on your suggestions. Wish you a smooth work ahead!
Reviewer 2 Report
Comments and Suggestions for Authors
1. Please read attached pdf file.
2. Have the dual-sensors proper aligned ?

Please check English writing style and grammer.
Author Response
Thank you for your revision suggestions on this paper. I have made corrections based on your feedback regarding areas where the expression was unclear or incomplete. Please review it again. Please see the attachment.

Reviewer 3 Report
Comments and Suggestions for Authors
I am not qualified to evaluate the English language of this manuscript, but it seems to me that the quality of the text needs to be checked in a number of places
Author Response
Thank you for your revision suggestions on this paper. I have proofread the text and made necessary corrections. Please review it.